# Sustainable Utilization of Waste Oyster Shell Powders with Different Fineness Levels in a Ternary Supplementary Cementitious Material System

Shanglai Liu [1], Yuan Wang [1], Bonan Liu [2], Zhen Zou [2], Yina Teng [3], Yidi Ji [1], Yubo Zhou [2], Lei V. Zhang [4],* and Yannian Zhang [2],*

1   College of Fisheries and Life Science, Dalian Ocean University, Dalian 116023, China; lsl19904019001@163.com (S.L.); wangyuan@dlou.edu.cn (Y.W.); jydidi@163.com (Y.J.)
2   School of Civil Engineering, Shenyang Jianzhu University, Shenyang 110168, China; m18342831904@163.com (B.L.); zh1377746456@163.com (Z.Z.); z15898274285@163.com (Y.Z.)
3   Changxin International Art School, Yunnan University, Kunming 650031, China; t15142587827@163.com
4   School of Civil and Transportation Engineering, Hebei University of Technology, 5340 Xiping Road, Tianjin 300401, China
*   Correspondence: lzhan666@uwo.ca (L.V.Z.); zyntiger@163.com (Y.Z.)

**Abstract:** As cement manufacturing accounts for 8% of global $CO_2$ emissions, there is an urgent need to tackle the environmental impacts of cement production and address the decarbonization of construction materials. Adopting supplementary cementitious materials (SCMs), including fly ash, slag, silica fume, etc., can be used as a partial replacement for ordinary Portland cement (OPC) to reduce $CO_2$ emissions related to the OPC industry, while providing benefits for waste valorization. This study aims to explore the sustainable utilization of a waste oyster shell powder (OSP)–lithium slag (LS)–ground granulated blast furnace slag (GGBFS) ternary SCM system in green concrete. The effect of OSP fineness on compressive strength, hydration products, pore structure, and transport properties in ternary SCM-based mortars was studied using a wide array of experimental techniques, including thermogravimetric analysis (TGA), scanning electron microscopy (SEM) analysis, Mercury intrusion porosimetry (MIP), the water absorption test and the rapid chloride penetration test (RCPT). The results revealed that the concrete with the ternary SCMs showed equivalent compressive strength compared to reference specimens. The water absorption and chloride ion charge of the RCPT in the concrete containing the ternary SCMs decreased by up to 30% and 81.4%, respectively. It was observed that the specimens incorporating the OSP with a mesh size of 3000 exhibited the highest compressive strength and the most refined microstructure.

**Keywords:** oyster shell powders; compressive strength; permeability; coupling effect; supplementary cementitious materials

## 1. Introduction

Millions of tons of shell waste are produced and abandoned in landfills in China every year [1]. These landfilled shells leach heavy metals from the heap due to dense emissions of land and air pollution during microbial decomposition and weathering. In some African countries, abandoned shells are difficult to handle, and millions of tons of abandoned shells pile up on shores and beaches each year. Open-air accumulation of abandoned shells is a potential habitat for microbes that attract organisms that are the carriers of potential diseases, causing public health problems [2]. The only effective solution to this problem is to recycle these shells as raw materials to develop new green building material [3]. In China, however, the utilization of discarded shells is less than 30%. Some scholars [4] have applied waste shells as concrete aggregate in the field of building materials. Whether they are used as coarse aggregate or fine aggregate, it is difficult for the crushing indices of burned waste shells to meet the standard of traditional sandstone, which greatly limits the application

of waste shells in the field of concrete. Shells contain a large amount of calcium oxide and calcium carbonate, which are two materials with a high recovery value [5]. Discarded shells are grinded into powder to treat highly acidic soils in agriculture [6]. In addition, calcium carbonate extracted from superfine OSP can be used to improve the properties of rubber and plastics [7]. Barros et al. [8] proposed a method to extract high purity calcium oxide from OSP as early as 2008. High purity calcium oxide belongs to high quality mineral resources and building materials. Akella et al. [9] showed that calcite and nepheline are the most abundant mineral crystals in OSP.

The accumulation of abandoned shells is increasing year by year, so it is unnecessary to worry about the shortage of raw materials. However, natural limestone belongs to the category of mineral resources. The increase in limestone consumption in the field of construction certainly means that a large number of mineral resources are mined. This has led to a series of problems, including environmental pollution, mineral resource depletion, and a series of collapses, landslides, and other security risks [8]. Therefore, it is urgently necessary to replace natural limestone with abandoned shells, a solid waste rich in calcium oxide, as a building material. According to the current literature, waste OSP rich in high quality calcium oxide has been used in the field of cement products [9].

Bassam et al. [10,11] investigated the possibility of replacing cement by grinding and burning bivalve clam seashells. Their results showed that the 5% replacement mix is the optimum percentage of replacement. However, these values are increased with higher levels of replacement. Seashell ash generally reduces workability. The ash also reduces concrete permeability after long periods of curing. Soltanza et al. [12] assessed the potential of waste shells as additives in mixed cement production, indicating that OSP can be used to develop mixed cement and improve the performance of concrete. Ali et al. [13] studied the effects of calcined oyster OSP as an additive on the hardening and microstructure properties of large volume slag cement. The results showed that the addition of calcined oyster OSP enhances the compressive strength of oyster shells at an early age, but excessive addition will lead to a decrease in their compressive strength. Abdelaziz et al. [14] conducted alkali activation treatment on OSP and prepared a new type of alkali-activated shell waste (AASW) mortar with a compressive strength of 22 MPa and a porosity of 16.5%. Hassan et al. [15] evaluated the effect of OSP on the chloride penetration of cement materials. OSP replacing part of cement can change the microstructure of materials and allow more ions to react with chloride through compaction, thereby reducing the corrosion risk of concrete reinforcement. From the existing research literature, it appears that waste OSP can be a perfect substitute for natural limestone powder. Therefore, OSP has the potential to develop into a supplementary cementitious material (SCM) [16]. Pusit et al. [17] used OSP instead of cement to prepare plaster mortar. Compared with traditional cement, the compressive strength was reduced during drying, but the shrinkage rate and thermal conductivity were lower. Although OSP has the potential to be developed as a SCM, the compressive strength deteriorates greatly with increases in the replacement rate. Therefore, OSP should be committed to the development of multiple cementitious systems.

Ground granulated blast furnace slag (GGBFS) and lithium slag (LS), two types of waste from the iron and lithium industries, are the most widely used types of slag to produce green concrete. Ali Shah et al. [18] developed environmentally friendly geopolymers using LS that showed better mechanical properties. He et al. [19] applied lithium slag to ultra-high performance concrete (UHPC). An appropriate LS content can improve the UHPC microstructure, and has a dense microstructure similar to the matrix in the interfacial transition zone (ITZ). LS is a promising silica fume substitute for the preparation of UHPC. Adding LS can also improve the mechanical properties of ordinary concrete, including its compressive strength, elastic modulus, drying shrinkage, and creep [20]. Granulated blast furnace slag (GGBFS) has a long history as a traditional SCM, which is very useful in the production of mortar and concrete, and can solve various problems, such as the workability, strength, and durability of concrete [21]. GGBFS is a by-product of blast furnace ironmaking, which contains a large number of active vitreous components and is a good precursor

of silica [22]. At present, GGBFS is more committed to the development of alkali-activated cementitious materials because of its water hardness [23]. Cao et al. [24] mixed GGBFS with ferronickel slag (FNS) to produce alkali-activated cement (AAC). When the proportion of GGBFS was reduced, the mechanical strength was greatly reduced, and the autogenous shrinkage, total porosity, and autogenous shrinkage were increased.

Although OSP as an SCM has been a well-studied topic, and although the efficacy of OSP in concrete is well documented and understood, there has been little research involving the effect of the fineness of OSP on the performance of SCM systems. Therefore, systematic studies are needed to better understand the effect of OSP in concrete SCM systems and clarify its performance, so as to recycle resources, reduce carbon dioxide emissions and achieve sustainable construction.

According to the existing literature, LS and GGBFS can theoretically cooperate with OSP to develop a ternary cementitious system. Therefore, this paper studied the compressive strength and permeability of a SP–LS–GGBFS ternary system with different fineness levels of OSP, and analyzed the influence mechanism of OSP fineness on the ternary system. In addition, the hydration degree and microstructure of the ternary system were studied by TG and SEM, and the pore structure was detected by Mercury intrusion porosimetry.

## 2. Materials and Methods

### 2.1. Raw Materials

The raw materials included OSP, GGBFS, LS, Portland cement (PO.42.5), standard sand, and water. PO.42.5 grade cement from Shenyang Shanshui Park Cement Co., Ltd. was used as a primary binder in this study. Oyster shells were provided by Rongcheng Xingyang Fish Flour Factory in Shandong Province, China. According to the research of Hyunsuk Yoon [25], the preparation process of OSP was cleaning, drying and calcination. The shells were washed with water to remove sundries on the surface, then dried at $105 \pm 5 \, ^\circ C$ for 24 h to remove any moisture content, and the OSP samples were then calcined for 2 h at 850–950 $^\circ C$, as recommended by Li [26]. The calcined shells then proceeded to the next step of grinding. Following Letwattanaruk et al. [27], OSP was used as a SCM, and the average particle size range should be from 0.5 to 40 μm. After grinding, a mesh size of 1250, 3000 or 6000 powder should be chosen. The raw material OSP was obtained after the above treatment. GGBFS was sourced from Jiyuan Steel Plant in Henan Province, China. LS was obtained from Tianyuan New Energy Materials Co., Ltd., located in Guangxi Province Qinzhou City, China. The chemical compositions of the three types of materials are shown in Table 1, the specific surface areas in Table 2 and the particle size distributions of OSP, LS, and GGBFS in Figure 1 via Brunauer–Emmett–Teller (BET) surface area analyses. The specific surface areas and particle size distributions of OSP under different mesh sizes are shown in Table 3 and Figure 2. The XRD in OSP was tested in temperatures ranging from 5° to 60°. Figure 3 shows that the main mineral composition of the OSP was dolomite, followed by quartz.

**Table 1.** Chemical composition and content of materials (mass fraction/%).

|       | SiO$_2$ | Al$_2$O$_3$ | CaO   | SO$_3$ | MgO   |
|-------|---------|-------------|-------|--------|-------|
| OSP   | 16.5%   | 0.3%        | 46.6% | 0.05%  | 36.3% |
| LS    | 54.5%   | 25.4%       | 6.4%  | 10.2%  | 0.6%  |
| GGBFS | 30.7%   | 15.9%       | 42.3% | 1.8%   | 6.7%  |

**Table 2.** The specific surface areas of the materials.

| Materials | OSP | LS | GGBFS |
|-----------|-----|-----|-------|
| Specific surface /m$^2 \cdot$kg$^{-1}$ | 2057 | 13,627 | 1206 |

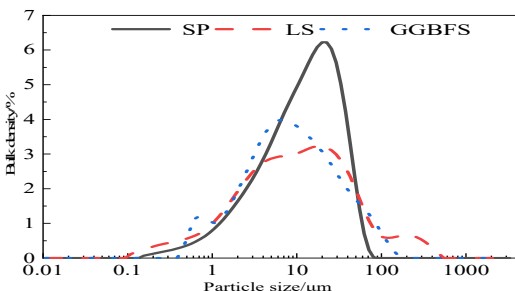

**Figure 1.** Particle size distributions of materials.

**Table 3.** The specific surface areas of OSP under different mesh sizes.

| Mesh size of OSP | 1250 | 3000 | 6000 |
|---|---|---|---|
| Specific surface /$m^2 \cdot kg^{-1}$ | 1624 | 2057 | 8479 |

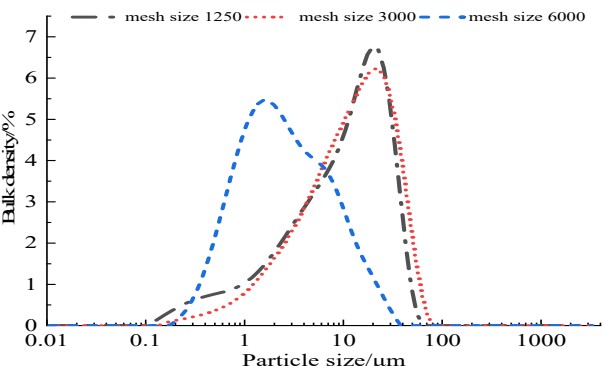

**Figure 2.** Particle size distribution of OSP under different mesh sizes.

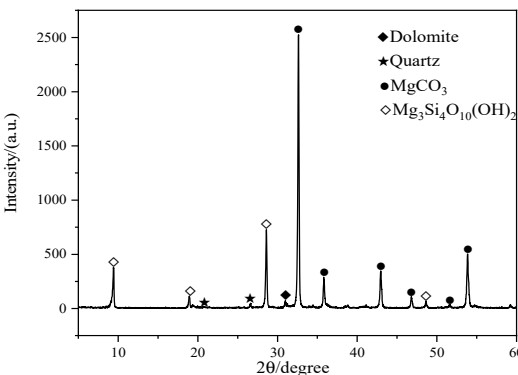

**Figure 3.** X-ray diffraction patterns of OSP.

### 2.2. Mix Design and Specimen Preparation

Table 4 presents the mixed details of the specimens. The required raw materials were poured into a JJ-5 planetary mixer to produce mortars as per China standard GB/T17671-1999 [28]. The mortars were placed in molds with a dimension of 40 mm × 40 mm × 160 mm and vibrated for 120 s. All specimens were demolded after 24 h and then cured in a standard curing room at 20 ± 2 °C with a relative humidity greater than 95%. Cubic 40-mm specimens were cast for measuring compressive strength. The purified slurry samples were crushed and soaked in anhydrous ethanol for 3 days to terminate hydration. The purified slurry samples were then dried in a vacuum drying chamber at 50 °C and further characterized. The P-1, P-2, and P-3 groups represented the fineness levels of different types of OSP. The mesh sizes were 1250, 3000, and 6000.

**Table 4.** Cement mortar ratios of groups.

| Serial Number | 0 | P-1 | P-2 | P-3 |
|---|---|---|---|---|
| Cement/g | 450 | 360 | 360 | 360 |
| OSP/g (1250 mesh sizes) | 0 | 45 | - | - |
| OSP/g (3000 mesh sizes) | 0 | - | 45 | - |
| OSP/g (6000 mesh sizes) | 0 | - | - | 45 |
| LS/g | 0 | 22.5 | 22.5 | 22.5 |
| GGBFS/g | 0 | 22.5 | 22.5 | 22.5 |
| Standard sand/g | 1350 | 1350 | 1350 | 1350 |
| Water/mL | 225 | 225 | 225 | 225 |

*2.3. Experimental Procedures*

2.3.1. Compression Tests

Compression tests were performed via GYE-300B universal testing machine (Beijing Kodak Jinwei Technology Development Co., Ltd. in Beijing, China). A loading rate of 2.4 kN/s, as stipulated by GB/T17671-1999, was used during the loading [28].

2.3.2. Thermogravimetric Analysis (TGA)

TGA was conducted using TA instruments (Q500 V20.13, USA) in a nitrogen gas flow with a 70 mL/min flow rate. Each testing sample was about 15 mg. The TGA was exposed to temperatures ranging from room temperature to 800 °C with a rate of 20 °C/min.

2.3.3. Permeability Tests

(1)    Water absorption

The test was carried out according to the British standard. The cylinder sample (U75 mm × 75 mm) was solidified and demolded. The standard curing times were 7 days and 28 days. The sample was dried in an oven (100 °C) for 3 days and cooled for 1 day. The sample was weighed, immersed in water for 24 h, and weighed again after 24 h. The average value of the two samples was the final result of each sample [29]. The hygroscopic rate was calculated according to Equation (1):

$$W = \frac{W_w - W_d}{W_d} \times 100\% \tag{1}$$

$W$ = hygroscopic rate; $W_W$ = weight of wet sample; $W_d$ = weight of dry sample.

(2)    Rapid chloride penetration test (RCPT)

The chloride ion penetration test was carried out according to ASTM C1202 [30], and the chloride ion charge was determined. The cylinder sample was 100 mm in diameter and 50 mm in length. After 28 days of water curing, the epoxy resin was smeared on the side, and the sample was fixed between two specific grooves. One side of the groove was connected with the sample, the other side was connected with the solution to connect the groove containing NaCl to the negative electrode, and the groove containing NaOH solution was connected to the positive electrode. The experiment started with DC 60V, and the total current that passed through the sample was measured for 6 h, which was automatically recorded every 5 min. The chloride permeability rating [31] is shown in Table 5.

**Table 5.** Chloride permeability rating.

| Chloride Permeability | Charge (Coulombs) |
|---|---|
| High | >4000 |
| Moderate | 2000–4000 |
| Low | 1000–2000 |
| Very low | 100–1000 |

2.3.4. Mercury Intrusion Porosimetry (MIP)

The porosity and pore size distribution were tested by the MIP experiment. The samples were selected after the compressive strength test. The samples were from the same cross section and the same depth, and the samples did not contain aggregate. After sampling, the samples were immersed in anhydrous ethanol for 7 days to terminate the hydration reaction. The samples were dried in a 60 °C oven and tested three days later. The relationship between the applied pressure and the cylindrical aperture was described by Washburn's equation.

2.3.5. Scanning Electron Microscopy (SEM)

The microstructure of the hydration products of the paste samples at 28 days was analyzed via SEM. All specimens were immersed in anhydrous ethanol for 3 days before the SEM test to prevent water and reaction, and then the samples were dried in the oven to remove anhydrous ethanol.

## 3. Results and Discussion

### 3.1. Compressive Strength Analysis

Figure 4 shows the effect of the fineness of OSP on the compressive strength of the ternary system. The fineness of OSP had a significant impact on the compressive strength of the ternary system at different ages. OSP as a SCM (P-3) caused the highest compressive strength in 3 days, which implies that the minimum fineness of OSP, in the absence of hydration reaction, has a microaggregate effect, thereby improving the compressive strength. The micro-aggregate effect plays a significant role in cementitious composites, which can compensate for the negative influence of pore structure defects on strength, as found by the study by G. Pan et al. [32]. However, with the increase in age, the micro-aggregate effect was gradually reduced. The increase in compressive strength at a later hydration stage was mainly due to the content and density of the C-S-H gel. The compressive strength of the P-2 stood out as the maximum at 7 and 28 days. The particle size distribution of OPS in the P-2 was different from P-3, which was concentrated in 1–10 μm rather than in 10–20 μm. The admixture particles of 10–20 μm can play a similar "nucleation effect" in the hydration process of cement, which helps to increase the content of C-S-H and make the hydration products uniformly distributed throughout the interface transition zone. Philippe et al. [33] also confirmed that this particle with a large specific surface area showed a better early strength. In the present study, when the specific surface area was slightly larger than the cement particle and the particle size distribution of the cement particle was similar, hydration was able to be better coordinated. This phenomenon shows that the fineness of OSP plays a vital role in the compressive strength contribution of the ternary system. In their review, Paul et al. [34] revealed that ultrafine powders act as fillers in cement-based materials, producing a dense matrix and reducing the growth of micropores. They also contribute to the formation of cement composite secondary reaction and to the improvement of strength. Conspicuously, in this study, the OSP of 6000 mesh sizes contributed the most to the early intensity. With the progress of hydration, the OSP of 3000 mesh sizes played the crystal nucleus effect of particles, showing the highest late strength. A similar effect is also reflected in the literature of Liu et al. [35].

### 3.2. Hydration Products Analysis

3.2.1. TGA

Figure 5 shows the TG results of each experimental group at 28 days. In each TG test, the first endothermic peak occurred between 100–120 °C. This can be attributed to the dehydration of the C-S-H gel. It is worth mentioning that in the TG image of the ternary system, there was no obvious endothermic peak formed by Aft dehydration near 200 °C, and only the slope changed, indicating that the Aft content in the ternary system was lower than that in the late hydration of pure cement paste. At 460–480 °C, there was a second endothermic peak. This was caused by the dehydration of CH by heat. At 690–730 °C,

there was a third endothermic peak. This was due to the decomposition of $CaCO_3$ formed by the carbonization of CH [36].

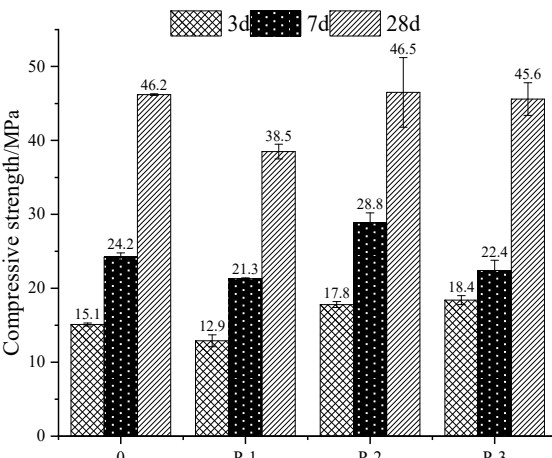

**Figure 4.** Compressive strength tests.

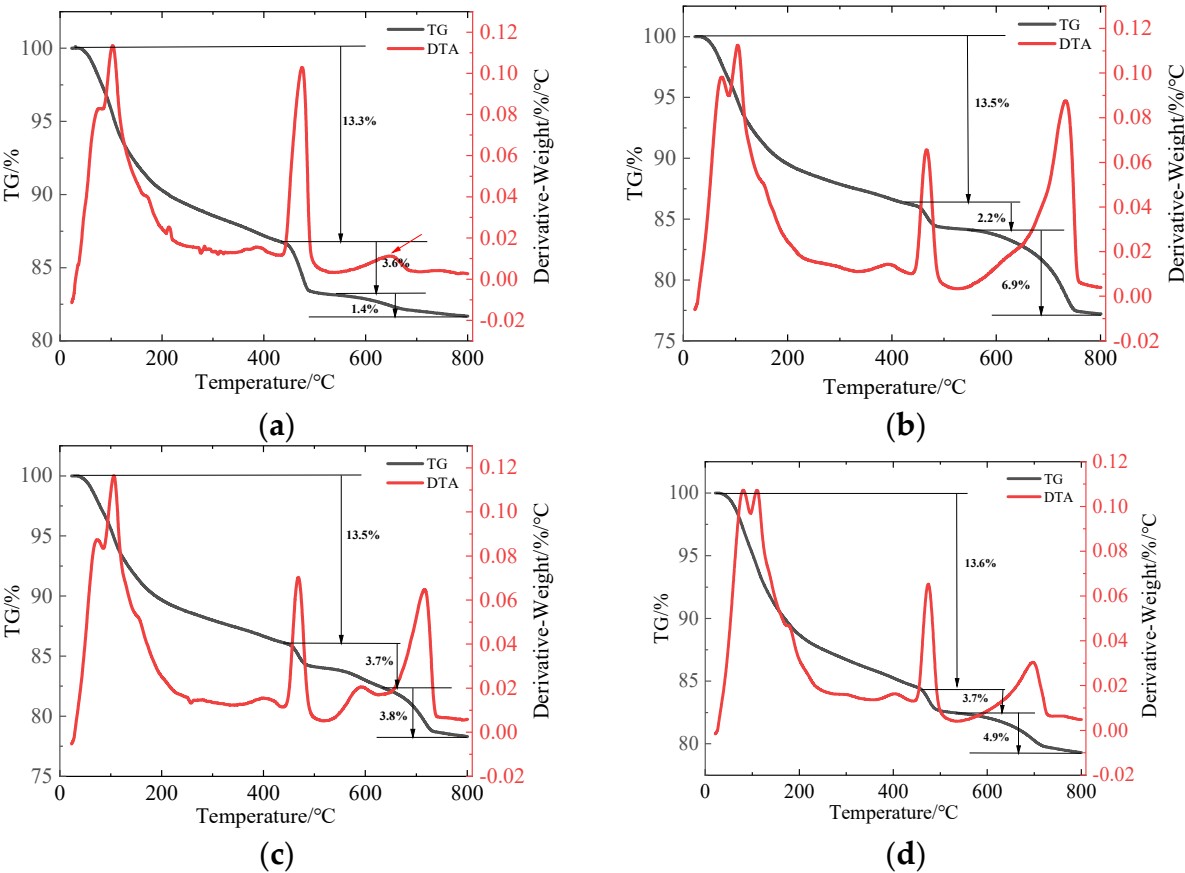

**Figure 5.** TG test result of each experimental group: (**a**) 0 group; (**b**) P−1 group; (**c**) P−2 group; (**d**) P−3 group.

In order to obtain the hydration process and hydration product content of the ternary system clearly, the C-S-H and CH contents of each group at 28 days were calculated by Equations (2) and (3) [37].

$$\text{CH} = WL_{\text{CH}} \times \frac{m_{\text{CH}}}{m_{\text{H}_2\text{O}}} + WL_{\text{CaCO}_3} \times \frac{m_{\text{CaCO}_3}}{m_{\text{CO}_2}} \tag{2}$$

$$H_2O = WL_{C-S-H} + WL_{CH} + WL_{CaCO_3} \times \frac{m_{H2O}}{m_{CO_2}} \times \frac{2}{3} \tag{3}$$

CH denotes the sample's relative calcium hydroxide content; $H_2O$ is the sample's relative water content; the mass loss of calcium carbonate caused by the removal of water by TG is denoted by $WL_{CH}$, %; $WL_{CaCO3}$ denotes the mass loss of calcium carbonate due to water removal by TG; $WL_{C-S-H}$; the molar mass of calcium hydroxide is denoted by $m_{CH}$; the molar mass of water is denoted by $m_{H2O}$; $m_{CaCO3}$ is the molar mass of calcium carbonate; $m_{CO2}$ is the molar mass of carbon dioxide.

$$CH = WL_{CH} \times \frac{74}{18} + WL_{CaCO_3} \times \frac{100}{44} \tag{4}$$

$$H_2O = WL_{C-S-H} + WL_{CH} + WL_{CaCO_3} \times \frac{6}{22} \tag{5}$$

The data from the TG images of the ternary system were substituted into Table 6 using Equations (4) and (5), as shown below:

**Table 6.** TG test of the yield of each substance.

| Serial Number | CH to Take Off the Water | Amount of CaCO₃ Decomposition | C-S-H Decomposition Quantity | H₂O Content | CH Content |
|---|---|---|---|---|---|
| 0 | 3.6% | 1.4% | 13.3% | 17.3% | 18% |
| P-1 | 2.2% | 6.9% | 13.5% | 17.6% | 24.7% |
| P-2 | 3.7% | 3.8% | 13.5% | 18.5% | 23.9% |
| P-3 | 3.7% | 4.9% | 13.6% | 18.6% | 26.4% |

Table 6 shows the $H_2O$ and CH contents in each experimental group. The CH content can reflect the degree of secondary hydration of SCMs. It is worth noting that when we introduced OSP, the content of CH in the product was dramatically improved. The hydration reaction of OSP was further confirmed, and the formation of CH provided a high alkalinity environment for the system to promote the secondary hydration of LS and GGBFS. The secondary hydration capacity of OSP with different fineness levels was also different. Gyu et al. [38] found that the fineness of limestone powder could affect the hydration rate and product. The CH content of P-2 was 23.9%, lower than that of P-1 and P-3. This indicates that P-2 had a profound degree of secondary hydration. Moreover, more C-S-H gel was observed by SEM. Klaartje et al. [39] pointed out a synergistic effect between the fineness of the materials and the fact that the more intensive grinding of fly ash reduced the gelatinization content. We suggest that OSP with a P-2 fineness can have a better synergistic effect with LS and GGBFS. Kan et al. [40] also proved that fineness and activity are not linear functions.

### 3.2.2. SEM Analyses

Figure 6 shows the morphology of two typical samples at the age of 28 days. It can be seen that there were many C-S-H gels and CH in the paste. Feng et al. [41] showed that substituting SCMs in cement can cause changes in the microstructure and chemical composition of the hydration products of mixed cement slurries. Hsu's study also showed that FA with different fineness levels had an impact on the properties and hydration products of colloids [42]. In the present study, most of the C-S-H gels adhered to CH and showed poor density and content, and many gel pores, as seen in Figure 6b. However, as can be seen in Figure 6a, there was an obvious weakness. This part of the weak area was attributed to the fact that the hydration products generated by cement and SCM are different in structure and composition and cannot form an organic combination. This weakness cannot be observed in Figure 6b. In Yakovlev's study, it was found that there was no strong adhesion of the hydration products when MWCNTs were mixed with CB paste,

while there was strong adhesion between MWCNTs and active silica powder (SF) [43]. One reason for this is that the resulting C-S-H gels were low, and the other is that finer OSP can fill this part of the pores and make the structure denser.

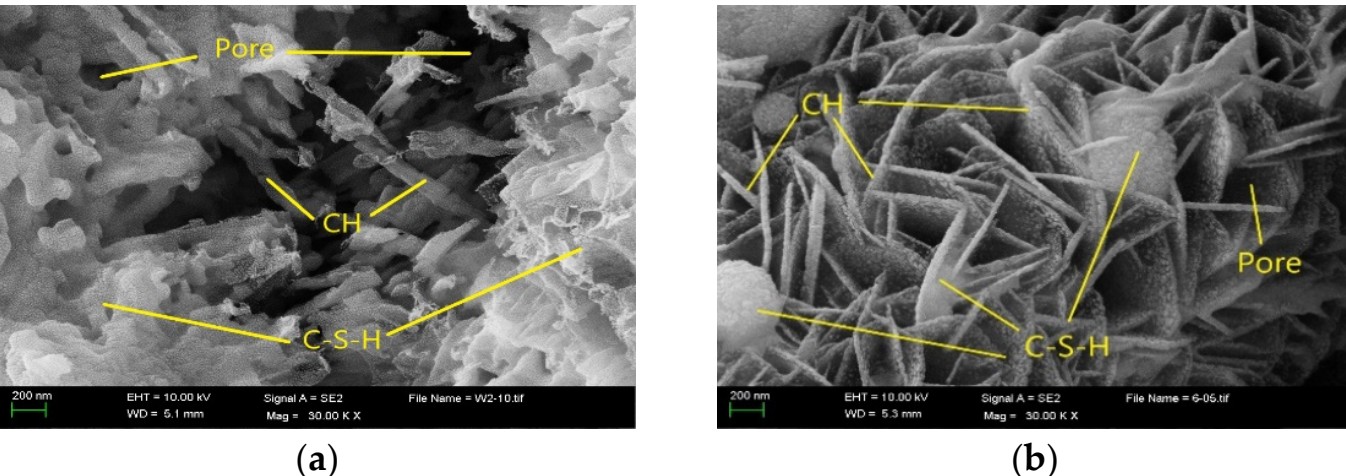

**Figure 6.** SEM images of hydration products: (**a**) P-2 group; (**b**) P-3 group.

### 3.3. Permeability and Pore Structure Analysis

### 3.3.1. Water Absorption and Rapid Chloride Penetration Test (RCPT)

Figure 7 shows the water absorption of mortar specimens at the age of 28 days. It was found that the water absorption declined when the mesh sizes of OSP were increased. In terms of fineness, the smallest OPS was able to fill the gap between the slurry and aggregate, thus preventing capillary water flow. Figure 8 illustrates the results of the RCPT of the specimens. In the RCTP test, we found the same rule. The introduction of the ternary system containing OSP reduced the total charge passing through, and the effect was particularly significant. The mortar formed a denser structure and reduced the penetration of chloride ions. This compact structure was less affected by the content of admixture. Khashaa et al. [44] also found such a pattern. This is due to the filling effect of unhydrated particles. Compared with the secondary hydration process, the filling effect dominates. In the present study, this filling effect was fully demonstrated when the mesh sizes of the shells changed. When the mesh size of OSP was 6000, the blocking effect on the chloride ion pathway was the best. The filling effect became more pronounced as the OSP mesh increased and the electric flux decreased.

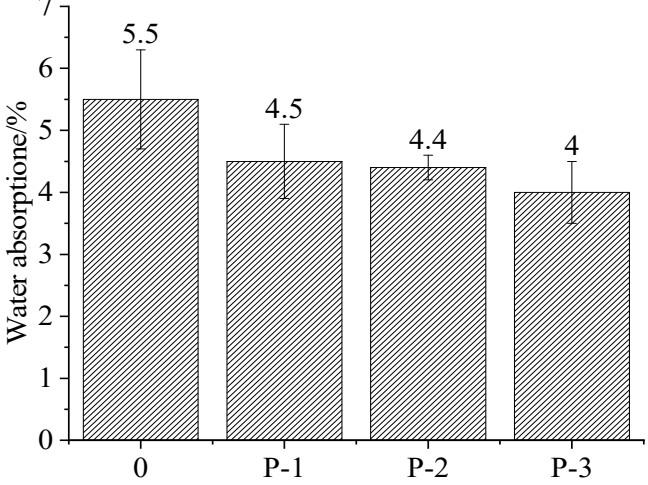

**Figure 7.** Water absorption of different groups.

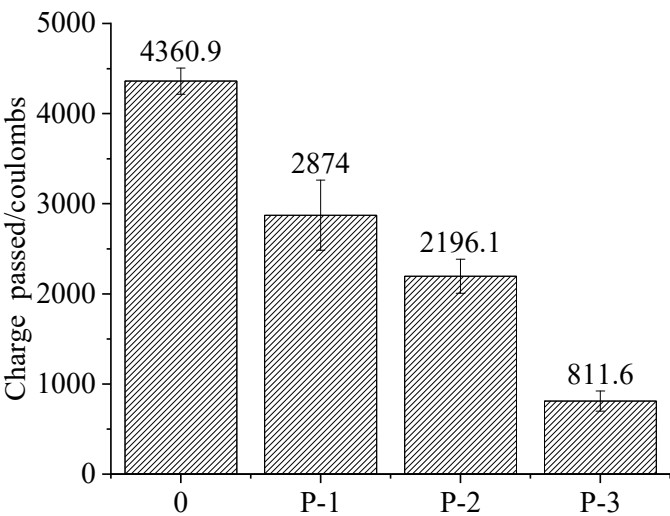

**Figure 8.** RCPT test results of different groups.

### 3.3.2. MIP Analysis and Pore Structure

Figure 9 compares the MIP curves of samples from the mixtures cured for 28 days. Table 7 shows the pore characteristic parameters. Compared with the experimental data, it is not difficult to find that P-3 had the best permeability, but the total porosity was higher than that of P-2.

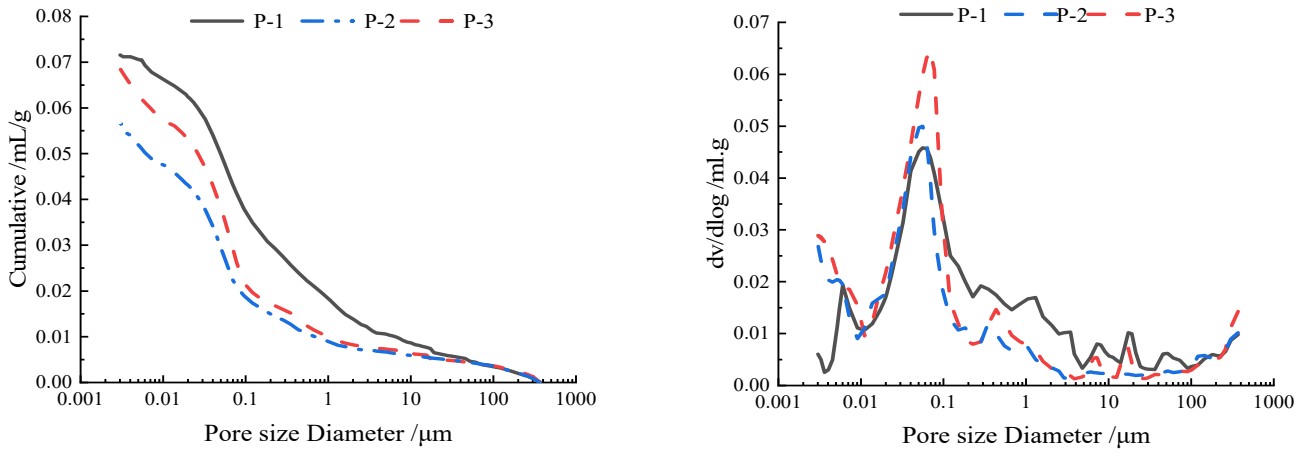

**Figure 9.** Pore size distributions in mortar samples at 28 days (mL/g).

**Table 7.** Pore characteristic parameters in mortar samples at 28 days.

| Serial Number | Total Pore Volume (mL/g) | Maximum Aperture/μm | Pore Size Distribution | | | |
|---|---|---|---|---|---|---|
| | | | 0–0.02 μm | 0.02–0.1 μm | 0.1–0.2 μm | >0.2 μm |
| 0 | 0.151 | 0.095 | 1.001 | 0.542 | 0.073 | 0.342 |
| P-1 | 0.151 | 0.055 | 1.002 | 0.56 | 0.099 | 0.488 |
| P-2 | 0.122 | 0.056 | 0.747 | 0.341 | 0.049 | 0.301 |
| P-3 | 0.143 | 0.069 | 0.908 | 0.428 | 0.055 | 0.329 |

An interesting phenomenon should be noted. According to previous studies [45], it is generally believed that the finer the particle size of the admixture, the better the filling effect and the better the pore structure. This meets the law in the case of single-component admixtures. However, particle fineness is not the only factor affecting pore structure in multi-component systems. In this study, the mesh size of 6000 OSP was the best, but the

porosity of the ternary system was higher than that of 3000 OSP. Therefore, more attention should be paid to the grading of particles rather than the fineness in the multivariate system. Guo et al. [46] introduced metakaolin to perform clearance grading for cement particles, and the results showed that it was more conducive to blocking the entry of chloride ions and preventing the release of binding chloride ions, thus improving the chloride ion resistance of cement-based materials. Figure 10 shows the aperture distribution. With the decline in OSP fineness, the pore size distribution was optimized, and the proportion of harmful pores decreased, while the proportion of harmless pores changed little, but the proportion of cementitious pores increased. Wang's study found that phosphorus slag with a specific surface area of 505 $m^2$/kg had higher reactivity than other fineness levels, resulting in a denser microstructure and significantly affecting porosity and pore size [47].

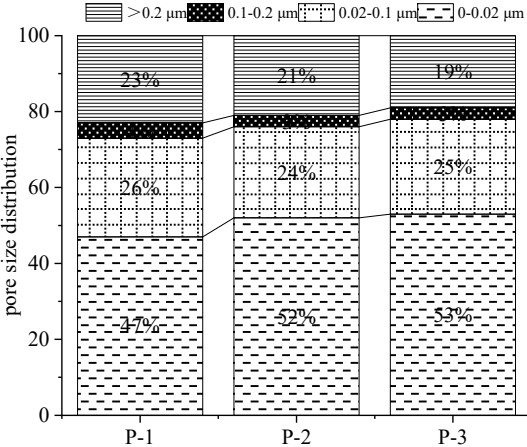

**Figure 10.** Pore size distribution in mortar samples at 28 days.

This result echoed that of the TG test. P-2 with better secondary hydration was able to produce more C-S-H gel and reduce the gel pore content. The smaller P-3 particles filled more larger pores through the micro-filling effect and reduced the content of harmful pores.

## 4. Conclusions

(1) The compressive strength of OSP (3000 mesh sizes) was 46.5 MPa in the ternary SCM composed of LS and GGBFS, which has the potential to develop into a high-performance SCM. There was no linear relationship between the compressive strength of the SCM and the fineness of OSP. Good particle size distribution was the key to the final hydration products.

(2) The introduction of the ternary SCM system enhanced the impermeability of mortar. The higher the OSP fineness was, the better the micro-aggregate effect. These fine particles can directly fill the mortar pores, thus enhancing impermeability.

(3) OSP with a mesh number of 3000 had the lowest total porosity. With the increase in OSP fineness, the pore structure distribution of mortar presented an optimization trend, and the proportion of harmful pores >0.2 μm gradually decreased. However, the fine OSP was not conducive to the secondary hydration of the ternary SCM system, resulting in an increase in the proportion of 0–0.02 μm gel pores.

(4) For OSP, CH can be provided in the hydration process of the ternary SCM system. When the size of OSP is too large, it is not conducive to the secondary hydration of SCMs, but the filling effect can make the mortar obtain better impermeability. Therefore, OSP with a mesh number of 3000 can be selected for better compressive strength, and OSP with a mesh number of 6000 can be chosen for better impermeability.

**Author Contributions:** S.L.: Methodology, Validation, Formal analysis, Investigation, Data curation, Visualization, Writing—original draft, Writing—review & editing. Y.W.: Validation, Formal analysis, Investigation, Data curation, Visualization, Writing—review & editing. B.L., Z.Z., Y.T., Y.J. and Y.Z. (Yubo Zhou): Methodology, Validation, Formal analysis. Y.Z. (Yannian Zhang): Conceptualization, Methodology, Validation, Project administration, Writing—review & editing. L.V.Z.: Conceptualization, Methodology, Validation, Project administration, Writing—review & editing. All authors have read and agreed to the published version of the manuscript.

**Funding:** The financial support provided by the National Key Research and Development Program of China (No.2019YFC1907202) and Major Science and Technology Project of Liaoning Province (No. 2020JH1/10300005) to the first author is gratefully acknowledged.

**Institutional Review Board Statement:** Not applicable.

**Informed Consent Statement:** Not applicable.

**Data Availability Statement:** The data of this study is available from the authors upon request.

**Conflicts of Interest:** The authors declare no conflict of interest.

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
