# Peer review of "Sustainable Utilization of Waste Oyster Shell Powders with Different Fineness Levels in a Ternary Supplementary Cementitious Material System"

_sustainability, doi:10.3390/su14105981_

Round 1

Reviewer 1 Report

This study aims to explore the sustainable utilization of waste oyster (OSP)-lithium slag (LS) -frosted blast furnace slag (GGBFS) ternary SCM system in green concrete.  The effects of OSP fineness on mortar compressive strength and permeability were investigated. Overall speaking, this is a very interesting study. The experiments were well-designed and the conclusions were well supported by the results. However, there are some problems which the authors should look into.

  1. OSP should be defined in the manuscript.
  2. The knowledge gaps should be explained, and the innovation of this study should be emphasized.
  3. The challenges of using waste materials in cementitious materials should be reviewed. In general, the chemical components and reactivity of raw materials sourced from different places vary widely. Therefore, intensive laboratory tests are usually required to evaluate the properties of materials. A recent study proposed the use of thermodynamic modelling to assist in the mix design of cementitious materials with waste byproducts. (Analytical investigation of phase assemblages of alkali-activated materials in CaO-SiO2-Al2O3 systems: The management of reaction products and designing of precursors. Materials & Design194, p.108975.)
  4. Did the oyster shell contain a lot of chloride which may lead to the chloride attack?
  5. The conclusions are too simple and more like the conclusions in a project report. More in-depth discussions should be provided, and the authors can talk about the perspectives and future studies.

Author Response

Table 1a: General Comments of Reviewer # 1:

Abbreviated Comment - Reviewer #1

Actions/ Corrections of Authors

This study aims to explore the sustainable utilization of waste oyster (OSP)-lithium slag (LS) -frosted blast furnace slag (GGBFS) ternary SCM system in green concrete.  The effects of OSP fineness on mortar compressive strength and permeability were investigated. Overall speaking, this is a very interesting study. The experiments were well-designed and the conclusions were well supported by the results. However, there are some problems which the authors should look into.

The authors would like to sincerely thank the editor for his valuable input and recommendations for improving the quality of the manuscript. And the questions raised by the censors have been supplemented and replied in the new manuscript.

Table 1b: Technical comments of Reviewer # 1:

Abbreviated Comment - Reviewer #1

Actions/ Corrections of the Authors

OSP should be defined in the manuscript.

Thank you for your suggestion. The meaning of OSP is redefined in the abstract and modified in the new text.

The knowledge gaps should be explained, and the innovation of this study should be emphasized.

Thanks to the examiner's careful examination, In The new manuscript we added "The effect of OSP fineness on The performance of ternary system SCMs is still unknown, especially in the aspects of hydration characteristics and permeability. In this paper,  the fineness of OSP is taken as a variable to explore the influence on the performance of ternary SYSTEM SCMs,  Which can provide a theoretical basis for a large number of applications of OSP in the future.

The challenges of using waste materials in cementitious materials should be reviewed. In general, the chemical components and reactivity of raw materials sourced from different places vary widely. Therefore, intensive laboratory tests are usually required to evaluate the properties of materials. A recent study proposed the use of thermodynamic modelling to assist in the mix design of cementitious materials with waste byproducts. (Analytical investigation of phase assemblages of alkali-activated materials in CaO-SiO2-Al2O3 systems: The management of reaction products and designing of precursors. Materials & Design, 194, p.108975.)

Thank you for your suggestion. For OSP in different regions, there are indeed differences in chemical composition and mineral composition, which is lacking in this paper. In subsequent studies, thermodynamic model will be used to assist in mixture design and composition difference study.

Did the oyster shell contain a lot of chloride which may lead to the chloride attack?

Thank you for this question. The material selected in this paper is not manual grinding of the initial oyster shell, but calcined oyster shell powder. The origin and parameters of these oyster shell powder are reflected in the text of the manuscript, so a large amount of chloride was not detected in this part of oyster shell powder.

The conclusions are too simple and more like the conclusions in a project report. More in-depth discussions should be provided, and the authors can talk about the perspectives and future studies.

Thank you for your suggestion. In the new manuscript, the conclusion of the article has been rewritten. And combined with the views of the inspectors to the future development of OSP put forward some guiding suggestions.

Reviewer 2 Report

The paper presents a good topic related to Sustainable utilization of waste oyster shell powders ternary supplementary cementitious material system with different fineness. The paper should be improved according to the comments in the attached file.

Author Response

Table 1a: General Comments of Reviewer # 2:

Abbreviated Comment - Reviewer #2

Actions/ Corrections of Authors

The paper presents a good topic related to Sustainable utilization of waste oyster shell powders ternary supplementary cementitious material system with different fineness. The paper should be improved according to the comments in the attached file.

The authors would like to sincerely thank the editor for his valuable input and recommendations for improving the quality of the manuscript. And the questions raised by the censors have been supplemented and replied in the new manuscript.

Table 1b: Technical comments of Reviewer # 2:

Abbreviated Comment - Reviewer #2

Actions/ Corrections of the Authors

The introduction contains many old references and many sentences without citations. it s good to cite these very recent and related papers in the introduction, literature  and to support your discussion referring to these papers.

(2020) Durability and mechanical properties of seashell partially-replaced cement. Journal of Building Engineering, 31: 1-10. https://doi.org/10.1016/j.jobe.2020.101328

(2019) Properties of concrete containing recycled seashells as cement partial replacement: A review. Journal of Cleaner Production. 237. DOI: 10.1016/j.jclepro.2019.117723.

Thank you for your suggestion. References to your recommendations have been cited in the introduction to the new manuscript, and older references have been replaced.

support these sentences with previous studies(2020). Effect of using mineral admixtures and ceramic wastes as coarse aggregates on properties of ultrahigh-performance concrete. Journal of Cleaner Production, 273, 123073.

Thank you for your suggestion. In the new manuscript, the literature recommended by the reviewer has been cited, and new literature has been added to support the narrative in the introduction.

What s JJ??

Thank you for this question. "JJ" refers to the model of mixer. In China, the preparation of cement mortar requires special equipment, and the model of mixer is "JJ-5".

Is there an unit??

The Unit is based on China Standard GB/T17671-1999, in which The preparation method of cement mortar specimen and The test method of strength are disclosed, which is The standard for The preparation and strength test of cement and cement mixture in China at The present stage.

Support by previous study, may you refer to 

(2012). Mechanical and permeability properties of the interface between normal concrete substrate and ultra high performance fiber concrete overlay. Construction and building materials, 36, 538-548.

Thank you for your suggestion. In the new manuscript, the literature recommended by the reviewer has been cited, and new literature has been added to support the narrative in the introduction.

Support the discussion by previous study

Thank you for your suggestion. Two articles were added to the new manuscript to prove the conclusions.

The resolution of Figs not clear

Thank you for your suggestion. In Fig5, the information repeated with the text description in each image has been removed, and the image has also been processed to obtain higher clarity.

Support the discussion by previous study

Thank you for your suggestion. Three articles were added to the new manuscript to prove the conclusions.

Support the discussion by previous study

Thank you for your suggestion. Two articles were added to the new manuscript to prove the conclusions.

Rewrite the conclusion to be more clear

Thank you for your suggestion. In the new manuscript, the conclusion of the article has been rewritten. And combined with the views of the inspectors to the future development of OSP put forward some guiding suggestions.

Reviewer 3 Report

The manuscript entitled ‘Sustainable utilization of waste oyster shell powders ternary supplementary cementitious material systems of oyster shell waste with different fineness’ is relevant for the Sustainability journal. It based on original research. The manuscript is well organized; however, it has one serious flaw as a lack of DISCUSSION. Moreover, it requires also some minor changes, such as:

  • Abstract: i) used research methods should be clarified; ii) the measurable results should be included; iii) please give the full name, when you implement the abbreviation the first time into the text.
  • Introduction: Could you support the data about the waste shells by the measurable data?
  • 1 Raw materials: What exactly is referred to by [23]? The process of cleaning? - Please, be precise.
  • Figure 5. Please give larger charts.
  • Discussion – lack of discussion part with comparison obtained result with up-to-date literature.

Author Response

Table 1a: General Comments of Reviewer # 3:

Abbreviated Comment - Reviewer #3

Actions/ Corrections of Authors

The manuscript entitled ‘Sustainable utilization of waste oyster shell powders ternary supplementary cementitious material systems of oyster shell waste with different fineness’ is relevant for the Sustainability journal. It based on original research. The manuscript is well organized; however, it has one serious flaw as a lack of DISCUSSION. Moreover, it requires also some minor changes, such as:

The authors would like to sincerely thank the editor for his valuable input and recommendations for improving the quality of the manuscript. And the questions raised by the censors have been supplemented and replied in the new manuscript.

Table 1b: Technical comments of Reviewer # 3:

Abbreviated Comment - Reviewer #3

Actions/ Corrections of the Authors

Abstract: i) used research methods should be clarified; ii) the measurable results should be included; iii) please give the full name, when you implement the abbreviation the first time into the text.

Thank you for this question. Based on the suggestions of the reviewers, we revised the abstract and responded to the reviewers' questions in the new manuscript.

Introduction: Could you support the data about the waste shells by the measurable data?

Thank you for this question. The introduction to the new manuscript, as suggested by the reviewer, adds measurable about OSP.

1 Raw materials: What exactly is referred to by [23]? The process of cleaning? - Please, be precise.

Thank you for this question. Refer to the method of reference [23] to process the material in this test.While the preparation of OSP was not clear in the previous manuscript, the process of OSP preparation is described in detail in the new manuscript.

Figure 5. Please give larger charts.

Thank you for your suggestion. In Fig5, the information repeated with the text description in each image has been removed, and the image has also been

Discussion – lack of discussion part with comparison obtained result with up-to-date literature.

Thank you for your suggestion. In the discussion part of the new manuscript, the literatures of recent years are added to further prove the thesis and experimental results of this paper. Meanwhile, some literatures in the introduction are replaced and added.

Round 2

Reviewer 2 Report

The authors responded to the comments. I recommend to accept the paper for publication

Reviewer 3 Report

The manuscript entitled ‘Sustainable utilization of waste oyster shell powders ternary supplementary cementitious material systems of oyster shell waste with different fineness’ have been improved, however it still requires implementation some changes, that was missed, such as:

  • Abstract: i) used research methods should be clarified; ii) the measurable results should be included.
  • Introduction: improper reference “Error! Reference source not found” (page 2, paragraphs 2 and 3) – it appears also in the other part of text.
  • Figure 5. Please give larger charts. It is still to small. The subscription of the charts are barely visible.
